# Time Intervals in the Pathway to Diagnosis of Patients with Cancer

**DOI:** 10.3390/nursrep15070261

**Published:** 2025-07-17

**Authors:** Joseba Rabanales-Sotos, Ángel López-González, Blanca Sánchez-Galindo, Gema Blázquez-Abellán, Juan Manuel Téllez-Lapeira, Jesús López-Torres-Hidalgo

**Affiliations:** 1Albacete Nursing School, University of Castilla-La Mancha, 02008 Albacete, Spain; joseba.rabanales@uclm.es (J.R.-S.); angel.lopez@uclm.es (Á.L.-G.); 2Albacete Integrated Care Management, Health Service of Castilla-La Mancha, 02008 Albacete, Spain; jmtellez@sescam.jccm.es; 3Albacete Pharmacy School, University of Castilla-La Mancha, 02008 Albacete, Spain; gemma.blazquez@uclm.es; 4Albacete Medicine School, University of Castilla-La Mancha, 02008 Albacete, Spain; jesusl@sescam.org

**Keywords:** early detection of cancer, primary health care, delayed diagnosis

## Abstract

**Background:** Around one-quarter of all people in the developed world die of cancer, with primary care being the main setting in which the disease is first suspected because the majority of patients consult a general practitioner (GP) when they present with symptoms. Diagnostic delay may thus be attributable to the patient, the GP, or the healthcare system. While some findings suggest that as much as half of the total delay consists of patient delay, more research is nonetheless needed into how GPs can facilitate access to diagnostic evaluation when patients experience symptoms. **Methods:** A retrospective observational study will be conducted to evaluate a cohort of patients diagnosed with cancer, with data being obtained from both primary and specialised care settings. Different time intervals will be analysed, dating from onset of first symptoms to diagnosis or initiation of treatment, and will be classified as: patient interval; primary-care interval; healthcare-system interval; diagnostic interval; treatment interval; and total interval. Study variables will include patient characteristics (socio-demographic, risk factors, morbidity, etc.), tumour characteristics (tumour stage, symptom onset, alarm symptoms, etc.), and healthcare characteristics (place of initial consultation, referral to specialised care, etc.). **Discussion:** The study will describe diagnostic delays in patients with cancer in primary care, considering the time elapsed between symptom onset and initial consultation, request for tests and/or patient referral, first evaluation in the hospital setting, and date of diagnostic confirmation and treatment initiation. Additionally, the study will make it possible to identify the patient-, healthcare-, and disease-related variables that intervene in the duration of such delays.

## 1. Background

### 1.1. Importance of Early Diagnosis of Cancer 

Around one-quarter of all people in the developed world die of cancer, and primary care is the main setting in which the disease is first suspected. Family history and risk factors are recorded, and primary prevention interventions to combat cancer are recommended [1]. Currently, population screening with a favourable risk/benefit balance [2] is solely available for breast, colorectal, cervical, and lung cancer (not on a generalised basis), but these represent only 29% of the total. Despite the fact that most patients with cancer still tend to be diagnosed when they already present with symptoms, early diagnosis is associated with better clinical outcomes and with better results according to the patients themselves [3].

General practitioners (GPs) have to assess the likelihood of cancer on the basis of a wide variety of signs and symptoms. According to a study conducted in the United Kingdom (UK), 80% of patients diagnosed with cancer consult their GP once or twice before being referred to a hospital [4]. UK general practitioners diagnose an average of one cancer per month in one of their patients [5]. The difficulty in diagnosis stems from the great variability in the clinical manifestations, depending on the type of cancer involved. The main challenge for GPs lies in maintaining a difficult balance between avoiding unnecessary interventions and not delaying actions required to address alarm symptoms [6]. Other factors that may be critical contributors to the difficulty in diagnosing cancer stem from differing patient beliefs, barriers in access to care, and some socio-demographic factors that contribute to delays in diagnosis.

There are few precise definitions of tumour-specific early symptoms in the literature [7], but the guidelines provided by NICE (National Institute for Health and Clinical Excellence) for suspected cancer are a reference guide for general practitioners, as they classify symptoms according to the level of urgency of referral. Cancer control requires public health interventions, and more resources will be needed for early detection in primary care to improve diagnosis and treatment at the specialist level [7].

To avoid delaying diagnosis, it would be desirable if patients with suspicion of cancer could undergo the pertinent tests via rapid diagnostic pathways [8]. Some countries have introduced a pathway for patients with cancer, often called the “rapid diagnostic pathway”, targeted at shortening the time elapsed between consultation and treatment in suspected cases of cancer [9]. Early diagnosis of cancer poses a challenge to GPs, and rapid diagnostic pathway referral should be the option if there is high suspicion of cancer. In a study in England [10], clinical suspicion on the part of the GP was identified as a strong predictor of cancer. Other predictors were patients’ age and having consulted the GP more than three times before referral.

A false concept is to assume that cancer is diagnosed at a hospital. Rather than diagnose, specialists tend to confirm (or rule out) cancer, since most cancers present with symptoms on which the GP is initially consulted [11]. It is likewise erroneous to assume that detection through screening renders symptomatic diagnosis any less relevant.

### 1.2. Duration of Diagnostic Intervals

There are a number of theoretical perspectives regarding the definition and measurement of time intervals in early diagnosis of cancer, with biomedical, psychological, or sociological approaches being predominant in each case. These are theoretical models based on identification of key events, which delimit different time periods. Although there is no widely accepted theoretical model, the Danish model developed by Olesen et al. in 2009 enjoys prominence [12].

Diagnostic difficulty is closely linked to the symptoms of presentation. The nature of these symptoms can critically influence the duration of the intervals between symptom onset and consultation of the GP, and between the latter and referral. Understanding which symptoms are associated with better time intervals is a research area of great interest. In England, a Department of Health Working Group [13] published the AARHUS statement, with recommendations about definitions and methodological approaches for designing and evaluating studies in this field. This statement contains definitions of the following time intervals between symptom onset and treatment initiation: (a) the interval attributable to the patient; (b) the interval attributable to primary care; and (c) the interval attributable to the healthcare system.

The initial symptoms are often common and non-specific, displaying a low positive predictive value (PPV), with an ensuing low likelihood of diagnostic accuracy. The decision to request tests, make a referral to the hospital, or “wait and see” can frequently be quite complex. NICE recommends referral to specialised care in cases where the PPV of the symptoms is higher than 3% [14]. A PPV of 5% should be considered highly predictive [8], assuming that 19 out of every 20 referred patients will not have cancer. To improve GPs’ diagnostic judgement, algorithms or computerised decision support systems that calculate the PPV for combinations of symptoms may well prove useful [15,16]. More research is currently needed to estimate PPV for symptoms whose importance is unknown and especially for combinations of symptoms and risk factors.

In the future, immunotherapy options will be supplemented with vaccines to prevent cancer. In recent years, circulating tumour cells (CTCs) and circulating tumour DNA (ctDNA) have received enormous attention as new biomarkers, affording the possibility of simultaneous detection of multiple primary cancers (MultiCancer Early Detection/MCED) [17]. Along with these advances, a complementary approach should also entail assessing primary care interventions to bring about the earliest possible presentation of cancer symptoms [18].

### 1.3. The Role of the General Practitioner

A study on Norwegian GPs [19] concluded that several factors were involved in raising awareness about cancer, including: (a) appropriate clinical knowledge about alarm symptoms, known as “red flags”, and about the epidemiology of each tumour; (b) knowing the patients well and being alert to changes in their appearance or behaviour; (c) intuition or presentment, which might be difficult to explain (this could be seen as the sum of acquired knowledge, clinical experience, and interpersonal relationships); (d) fear of diagnosis of cancer, regarding not only the patient’s fear of having the disease but also the physician’s fear of overlooking it. In the Norwegian study, a considerable number of GPs used the word “uncertainty” in relation to decision-making on the basis of suspicion of cancer.

GPs are directly involved in the initial diagnosis of more than 85% of all cases of cancer [5]. A Danish cohort study targeted at identifying variables associated with a longer delay in diagnosis [20] found that this involved a high degree of healthcare pressure and little prior knowledge of the patient.

GPs’ response to suspicion of cancer should be to arrive at an appropriate diagnosis in collaboration with the hospital, and the necessary radiological and endoscopic procedures or other techniques should be made available. However, in some healthcare systems, GPs have very limited access to these procedures and instead act as “gatekeepers”, needing to refer almost everything to the hospital, something that could contribute to the delay in diagnosis. A study of 19 European countries [21] found that “gatekeeper systems” register a lower survival rate at 1 year. There can be no doubt that delays in diagnosis, something that has unfortunately become an accepted tradition in many health services, are an important contributory factor.

It is not a matter of investigating all the symptoms at the initial consultation, but it is advisable to explain the uncertainty surrounding the cause of symptoms to patients, ensuring that they understand when they should return for another consultation. Following up with patients over time can be a good approach for non-specific symptoms. NICE includes recommendations in this regard [22], such as offering low-risk patients a review of symptoms within an agreed period and being alert to the possibility that the test results may be false negatives.

Our hypothesis is that in primary care, the diagnostic delay in cancer patients is high, and we consider that this is conditioned by the characteristics of the patient, the organisation of the health services, and the characteristics of the tumour. We consider that, despite the fact that screening is capable of diagnosing the disease during the pre-symptomatic phase, a high proportion (over 50%) of breast, colorectal, and cervical tumours are identified because patients already have symptoms and consult their family doctor. Finally, we consider that the proportion of tumours that debut with warning signs is low and that there is a large variability in the forms of cancer presentation. The main aim of this study is, thus, to describe diagnostic delay in patients with cancer in primary care, considering: (a) the time elapsed between symptom onset and initial consultation; (b) the time elapsed between initial consultation and request for tests and/or referral to hospital; (c) the time elapsed between referral and first evaluation in the hospital setting; (d) the time elapsed between first evaluation in specialised care and date of diagnostic confirmation; and (e) the time elapsed between diagnostic confirmation and treatment initiation. The secondary objectives are: (1) to identify the variables of the patient (age, sex, morbidity, etc.), organisation of healthcare services (specific referral circuits, screening programmes, priority of referral, etc.), and diseases (site, stage, etc.) that are involved in the delay times; (2) to describe the form of tumour presentation in primary care, quantifying the alarm symptoms described in each tumour (NICE Guidelines); and (3) to ascertain the proportion of tumours of breast, cervical, and colorectal cancer that are not diagnosed through screening programmes.

## 2. Methods

### 2.1. Study Design and Scope

A retrospective observational study will be conducted to evaluate a cohort of patients with cancer, with data being obtained from the date of symptom onset to that of treatment initiation. For study purposes, data pertaining to both the primary- and hospital-care settings will be needed. The patients will belong to three basic health areas of the city of Albacete, a city of 174,137 inhabitants that is located in the southeast of Spain and belongs to the autonomous community of Castilla-La Mancha. Patients will be drawn from three basic health areas in the city of Albacete, situated in the Castile-La Mancha Region in the southeast of Spain.

### 2.2. Sample Size and Selection of Participants

Based on morbidity lists sourced from computerised primary-care clinical histories, all patients diagnosed with cancer across the period 2015–2024 will be selected (inclusion criterion). Patients will be identified using the International Classification of Diseases, 9th Revision (ICD-9), and/or International Classification of Primary Care, 2nd edition (ICPC-2) codes. The following exclusion criterion will be applied: any tumour not classifiable as a malignant neoplasm (i.e., any tumour that does not invade nearby or other distal tissue via the blood or lymphatic system).

An estimated total of 3054 patients were diagnosed with cancer in the three health areas across the study period. This sample size will make it possible to compare groups of patients by estimating relative risks (RR) of 2.0 or more in cases where the outcome risk in the least exposed group is 2%, assuming alpha = 0.05 in a two-sided hypothesis and a power of 90%. Likewise, this sample size will make it possible to calculate proportions with a 95% confidence level and a minimum precision of ±1.77%.

### 2.3. Study Variables

Dependent variables:


-“Patient interval”: time elapsed between detection of first symptoms and initial consultation.-“Primary care interval”: time elapsed between initial consultation and referral to hospital (includes the “physician interval” or time elapsed between initial consultation and first test requested in primary care, in those cases where this has been requested).-“Healthcare interval”: time elapsed between first test requested by the GP and treatment initiation (includes “hospital care interval” or time elapsed between referral and treatment initiation).-“Diagnostic interval”: time elapsed between initial consultation and diagnostic confirmation.-“Treatment interval”: time elapsed between diagnostic confirmation and treatment initiation.-“Total interval”: time elapsed between symptom onset and treatment initiation.


Independent variables:


-Patient characteristics:
-Age and sex.-Risk factors for each tumour (classification of risk factors described by Marzo-Castillejo) [23], including age, family history, toxic habits, obesity, reproductive history, hormone treatment, etc.-Patient’s health problems classified as per the ICPC-2 or ICD-9.-Tumour characteristics:-Tumour type and stage (TNM classification).-History of cancer in first-degree blood relatives.-First tumour-related symptoms, including alarm symptoms (NICE Guidelines, 2015, updated in 2023).-Healthcare characteristics:-Place of consultation for first symptoms (health centre, emergency service, etc.).-Request for supplementary tests by the GP.-Patient’s referral to specialised care, department to which he/she is referred, and priority of referral (standard/preferential/urgent).-Visits to the GP in the year preceding diagnosis and the number of consultations between symptom onset and referral.-Participation or non-participation in screening programmes.-Use of specific referral circuits.


### 2.4. Data Collection and Data Sources

Both the selection of patients and collection of the necessary data will be undertaken at the participating health centres (Albacete Areas IV, V-B and VIII). The research team includes GPs and family medicine interns allocated to these hospitals, all with access to patients’ clinical histories.

The data source will be the clinical history, from which the necessary information will be obtained for study purposes, including specialised care reports. All the data collected will be included in a study-specific electronic case-report form. This will be drawn up in the following stages: selection of the necessary data after the literature review; definition of codes; design of the questionnaire format; and drafting of an instruction manual.

An initial pilot test will be performed to check that all the data can be obtained and that the instructions contained in the case-report form are fit for purpose, and to record the time required for its completion. A training session will also be held to ensure standardisation in the collection of data by the researchers.

### 2.5. Strategy of Statistical Analysis

The information contained in the database will be processed and then analysed using the IBM SPSS statistics programme (Table 1). The strategy of analysis will be to describe the variables by using proportions and the construction of 95% confidence intervals, or alternatively, by using measures of central trend and dispersion according to the nature of the variables. A comparative analysis will be made of the respective time intervals in diagnosis of cancer in subjects with different characteristics, using tests of comparison of means (Student’s *t* and ANOVA) or, where necessary, non-parametric tests (Mann-Whitney U and Kruskal-Wallis H), with a significance level of *p* < 0.05. Bonferroni Correction analysis will be used to address the problem of multiple comparisons.

Variables that display a statistically significant association with the time intervals in the diagnosis of cancer will be included in multiple linear regression models to study the dependency relationship of these time intervals, identifying the existence of confounding or interaction factors. Missing data will be quantified, and their pattern will be analysed. In order not to introduce bias, cases with insufficient information will be excluded from the analysis. Interpretation of the model will be determined by the statistical significance and value of the coefficients of the independent variables as an expression of their contribution to the variability of the dependent variables.

Survival analysis will be used to describe patients’ progress across the main time intervals by calculating the likelihood of being kept waiting to be diagnosed or receive treatment at different points in time (every 3 months) together with the corresponding confidence intervals. As a technique of analysis, the actuarial method of estimation will be applied using the survival procedure of the SPSS system, which enables survival times to be grouped in regular intervals. Survival curves will be calculated using the Kaplan-Meier method, with the cumulative estimated likelihoods of survival being plotted by reference to time (the likelihood that a patient will be waiting to be diagnosed or treated in each period considered). The survival curves will then be compared with the Mantel-Haenszel test (log-rank).

Lastly, Cox proportional hazards models will be fitted (Coxreg procedure of the SPSS system) to analyse the effects of the independent variables on the dependent variable “waiting time until diagnosis or treatment initiation”. These models will make it possible to establish prognostic factors significantly related to waiting time. The overall significance of the model will be evaluated with Rao’s test, as well as each variable with the likelihood ratio test.

### 2.6. Ethical Aspects

This is a research project approved by the Research Ethics Committee of the Albacete University Teaching Hospital Complex. Pursuant to Regulation (EU) 2016/679 of the European Parliament and of the Council of 27 April 2016, on the protection of natural persons with regard to the processing of personal data and on the free movement of such data (General Data Protection Regulation/GDPR), the personal data required will be those required to meet the designated objectives. Neither the name nor the identity of study subjects will appear in the database or in any of the reports, as they were anonymized at the time of extraction of the necessary data. These and other related aspects will be undertaken and performed in accordance with the provisions of the 2018 Personal Data Protection & Digital Rights Guarantee Act (Ley Orgánica 3/2018, de 5 de diciembre, de Protección de Datos Personales).

The study complies fully with clinical best practices and the basic principles laid down in the Helsinki Declaration.

## 3. Discussion

Physicians’ compliance with record keeping in terms of patients’ clinical histories is likely to vary widely, often displaying missing or incomplete information about delay intervals in the diagnosis of cancer and other study variables. The project plans to obtain information from both primary care and hospital records in order to cross-check information from different sources and, thus, increase the validity and reliability of the results. It will be especially difficult to pinpoint both the date of symptom onset and the initial consultation on these symptoms in a high proportion of patients. In other cases, data will be incomplete or lacking due to the fact that these are patients who initiated their diagnostic study in other health services or in private clinics.

To prevent bias in the measurement of time intervals, attention will be paid to the recommendations contained in the AARHUS Statement [13], which includes a useful checklist on the design of these types of studies. In line with these recommendations, the healthcare context in which the study is based has been fully described, and an attempt has been made to define the time intervals correctly, particularly the initial and final points of each.

The results will make it possible to quantify the diagnostic delay in cancer patients in primary care and to describe the different time intervals previously described in the literature that are considered useful, from the appearance of the first symptoms to diagnostic confirmation and the start of treatment. In addition, the results of the study will make it possible to identify variables that intervene in these intervals, both those that depend on the patients and those that depend on the characteristics of the tumour or the organisation of the health services. Furthermore, the project will provide information on how tumours are presented in primary care and will make it possible to quantify the alarm symptoms in each tumour. If these intervals prove to be high, the study will provide useful information for proposing greater accessibility to diagnostic tests for GPs in the study area, and the implementation of preferential referral circuits that will help to improve early detection of cancer and, by extension, prognosis of the disease.

It is likely that the results will serve to reduce GPs’ uncertainty when it comes to making decisions on the basis of suspicion of cancer. Furthermore, these results may also be of use in other health areas seeking to optimise the delay in the diagnosis of cancer.

## Figures and Tables

**Table 1 nursrep-15-00261-t001:** Summary table of the planned statistical analysis.

Goal of the Analysis	Statistical Test/Model	Statistical Software
Descriptive analysis	Description using proportions and construction of 95% confidence intervals, as well as measures of central tendency and dispersion	IBM SPSS statistics v. 28.0






Comparative analysis of time intervals between subjects with different characteristics	Comparison of means tests (Student’s *t*-test and ANOVA) and non-parametric tests (Mann-Whitney U and Kruskal-Wallis H)	



Relationship of time intervals to independent variables	Multiple linear regression models	


Evolution of patients in the main time intervals (probability of remaining awaiting diagnosis or treatment)	Survival analysis. Actuarial and Kaplan-Meier estimation methods. Comparison of survival curves with the Mantel-Haenszel (Log Rank) test	


Effect of independent variables on waiting time to diagnosis or to initiation of treatment	Cox proportional hazards models	

## Data Availability

Not applicable.

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
