# Peer review of "Time Intervals in the Pathway to Diagnosis of Patients with Cancer"

_nursrep, 2025, doi:10.3390/nursrep15070261_

Round 1

Reviewer 1 Report

Comments and Suggestions for Authors

The introduction is comprehensive and well organized, offering a logical progression from the importance of early cancer diagnosis to the role of general practitioners and the study’s specific aims. The research design is clearly outlined, including details on sample size, variable identification, and planned statistical analyses. However, the introduction could be improved by consolidating repetitive points, reducing less relevant content—such as extended discussion of future diagnostic innovations and international system comparisons—and focusing more tightly on the context of primary care. In the methods section, please clearly distinguish between dependent and independent variables, and specify the risk factors for each tumor type included in the analysis. Adding a summary table that links each analysis goal with its corresponding statistical test/model and software will further clarify your methodology. Additionally, greater attention is needed to the patient perspective and the specificity of the research gap; explicitly discuss how patient beliefs, access barriers, and sociodemographic factors contribute to delays, and clearly outline how this study adds to existing knowledge, especially in specific healthcare settings. Please see the specific line-by-line suggestions for improving clarity and reducing redundancy in the manuscript.

Reviewer 2 Report

Comments and Suggestions for Authors

The study seem to be key in contributing to the aspect of delay in cancer field. I suggest perhaps alignging the discussion to what the study aims to improve. .It was a bit confusing to align the discussion with the bacground. Could not make claer indications of the use of statistics . 

Comments on the Quality of English Language

 Suggest  make use of an English language editor could help bring out the main contribution of the study

Reviewer 3 Report

Comments and Suggestions for Authors

This protocol paper outlines a retrospective observational study aimed at quantifying diagnostic delays in patients with cancer within the Spanish primary care system. Understanding these delays is critical for identifying systemic and patient-level bottlenecks and improving early detection pathways, ultimately enhancing patient outcomes and healthcare efficiency.

Major Comments:

Title and throughout the manuscript: Please avoid the phrase “cancer patients.” Instead, use “patients with cancer” to reduce stigmatizing language.

Page 1-2: The introduction provides a good background, but the rationale for choosing Albacete as the study setting could be further clarified, especially regarding representativeness.

Page 4, Line 165: The hypothesis is clearly stated but could benefit from reframing as a testable statement or specific research question.

Page 7, Lines 281–293: The protocol mentions GDPR compliance. It would be helpful to briefly mention whether data will be anonymized at the point of extraction.

Page 7, Lines 295–301: Given the retrospective nature, potential biases due to incomplete documentation are acknowledged. However, it is advisable to outline mitigation strategies more concretely (e.g., triangulating data from hospital and primary care records).

Minor Comments:

Throughout the manuscript: 

  • Ensure consistency in terminology. For example, use either "fast track" or "rapid diagnostic pathway" but avoid alternating terms without clarification.
  • The manuscript is generally well-written. However, minor grammatical polishing would enhance readability (e.g., streamline complex sentences such as Page 3, Line 100–102).

Page 6, Line 250–275: Consider clarifying how missing data will be handled in regression and survival analyses.
